# First Case Report of Successful Treatment of *Mycobacterium abscessus* Infection in a Cat in Thailand

**DOI:** 10.3390/ani15070925

**Published:** 2025-03-23

**Authors:** Thapanee Chuenngam, Suttiwee Chermprapai

**Affiliations:** 1Dermatology Center, Kasetsart University Veterinary Teaching Hospital, Faculty of Veterinary Medicine, Kasetsart University, Bangkok 10900, Thailand; fvettnc@ku.ac.th; 2Department of Companion Animal Clinical Sciences, Faculty of Veterinary Medicine, Kasetsart University, Bangkok 10900, Thailand

**Keywords:** cats, MALDI-TOF MS, mycobacteria, non-tuberculous mycobacteriosis

## Abstract

Mycobacterial infection, caused by acid-fast bacilli, is an important infectious disease of global health concern for humans and animals. In cats, these infections can lead to various conditions, including feline tuberculosis, feline leprosy syndrome, and non-tuberculous mycobacteriosis, which often arise from skin contamination due to trauma or surgery. Successful treatment is challenging and requires a combination of two or three antibiotics in most cases. Prognosis varies widely depending on the mycobacterial species and disease extent.

## 1. Introduction

Mycobacterial infection is an important infectious disease of global health concern for humans and animals [1,2,3,4]. The causative agent is a Gram-positive, aerobic, acid-fast, bacilliform mycobacterium species [2,3,5], having a complex taxonomy. Some species are serious zoonotic pathogens, while many mycobacterial species can infect cats, leading to different clinical presentations [1,2,5]. These species have the ability to resist environmental conditions and external factors [2,3,4]. In cats, the mycobacterial disease can cause different syndromes, including tuberculosis (TB), feline leprosy syndrome (FLS), and non-tuberculous mycobacteriosis (NMT) [1,2,4,5,6].

NMT consists of a large number of mycobacterial species that can be classified into many groups based on growth ability in culture (slow-growing, rapid-growing, or non-growing) and on pathogenicity properties (absolute, facultative, or saprophyte pathogens) [2,3,5,6]. The skin is the primary source of infection, typically through traumatic or surgical wounds, contaminated with mycobacteria; soil, water, and decaying vegetation are the main sources of this agent [1,2,5]. It is replicated in lipid-rich tissues, such as skin at the ventral abdomen and inguinal areas, leading to cutaneous panniculitis characterized by multiple punctate draining tracts and subcutaneous nodules [1,2,4,5,6,7]. The prognosis of NTM is very variable, being good to poor, depending on the extent of disease and the mycobacterial species involved [2,5].

Treating a potential case of feline mycobacteriosis is challenging as it is almost always long-term and requires a combination of two or three antibiotics in most cases [2,5].

This case report describes the diagnosis and successful treatment of non-tuberculous mycobacteriosis in a cat. Based on the literature, this is the first published case report of successful treatment of *M abscessus* infection in a cat in Thailand.

## 2. Case Description

*Pre-diagnosis*. A female, neutered, domestic shorthair cat aged 2 years was presented with a 1-month history of subcutaneous nodules with multiple sites of draining tract at the right ventral flank, left inguinal, and prepubic area (Figure 1), which occurred 2 months after fighting with a stray cat. The cat was initially diagnosed with a fungal infection and treated with oral itraconazole for 2 weeks at the previous clinic without any clinical improvement.

*Diagnosis.* In the current treatment regime, on the first visit, a complete blood count showed mild normocytic normochromic anemia with a hematocrit of 24.4% (reference interval [RI] 30–50%) and a normal serum chemistry panel. A discharge from the draining tract was collected for fungal culture, bacterial culture, and susceptibility analysis. A fine-needle aspiration of the granuloma area was taken for cytology. The biopsy of the lesion was declined by the owner due to the need for anesthesia. Amoxicillin–clavulanic acid at 19 mg/kg PO q12h was prescribed before the results of the laboratory report. The cytology revealed multinucleated giant cells, macrophages, and large spindle cells, as well as degenerate neutrophils, indicating pyogranulomatous inflammation (Figure 2). There was no fungal growth after 14 days of culture. Bacterial culture of the draining tract yielded the growth of *Mycobacterium abscessus*, one of the mycobacterial species that cause NTM, which was identified by the matrix-assisted laser desorption-ionization-time-of-flight mass spectrometry (MALDI-TOF MS) method. It was susceptible to amikacin, azithromycin, tetracycline, and imipenem, with intermediate susceptibility to doxycycline and erythromycin and resistance to many other drugs (Table 1).

*Treatment and evolution.* Based on the susceptibility results, azithromycin at 10 mg/kg PO q24h (Azith^®^, Siam Bheasach Co., Ltd., Bangkok, Thailand) and topical amikacin sulfate 5% gel (Likacin^®^ gel 5%, S Charoen Bhaesaj Trading, Bangkok, Thailand) were applied inside the lesions once daily to control the infection. After 36 days of treatment, the subcutaneous nodular lesions showed a mild decrease in size but still had mucoid discharge from the opening of the draining wound (Figure 3). Then, doxycycline at 8 mg/kg PO q24h (Siadocin^®^, Siam Pharmaceutical Co., Ltd., Bangkok, Thailand) and lipopolysaccharide (LPS^®^, T.J. Animal Health Co., Ltd., Samut Prakan, Thailand) immunostimulant were combined in 1 tablet a day PO from this day. The lesions gradually improved and showed complete improvement after 14 weeks of the treatment (day 98 of azithromycin and day 61 of doxycycline), as shown in Figure 4. Hematology and blood chemistry analyses were performed during follow-ups, revealing mild anemia and normal serum biochemistry values. The cat displayed normal vital signs and behavior. At 8 months after the complete recovery of the lesions, the cat had not had a relapse from the infection.

## 3. Discussion

*Mycobacterium abscessus* belongs to the group of rapid-growing, non-tuberculous mycobacteria (NTM), which are diverse and ubiquitous in the environment [8]. In cats, rapid-growing NTM infections have been reported in the USA, Australia, the United Kingdom, Canada, Western Europe, New Zealand, and South America [8]. In Asian countries, there have been no reports of rapid-growing NTM in cats, except for one report of *Mycobacterium intracellulare*, a slow-growing NTM, in 2023 [9]. Therefore, this present case is the first reported in Thailand.

NTM primarily causes cutaneous lesions in cats, characterized by multiple punctate draining tracts and subcutaneous nodules [1,2,4,5,6,7,8,10]. Typically, infection occurs through direct inoculation into subcutaneous adipose tissue via contaminated soil, plant material penetrating wounds, injuries from fighting, or surgical wound contamination [2,8]. Diagnosing mycobacterial infection in cats is difficult and challenging due to the different zoonotic risks and varying sources of mycobacteria [1,5,11], each with different prognoses and antibiotic responses. Appropriate sample collection for mycobacterial detection is crucial. Discharge or tissue from an infected area should be collected for cytology and histopathology with acid-fast (e.g., Ziehl-Neelsen) staining. Granulomatous inflammation with foamy macrophages containing variable numbers of acid-fast bacilli is a typical finding [2,5,8]. Specialized laboratory culturing with antimicrobial sensitivity testing or molecular diagnostics is necessary to identify mycobacteria at the species level [2,5]. In the past decade, there have been some reports on the use of the MALDI-TOF MS method to identify NMT species [10,11,12,13,14,15]; this technique is a powerful tool for the routine identification of bacteria, yeast, and molds. The identification is based on species-specific spectral fingerprints obtained from crude extracts of whole cells [10,11,12,13], which is an accurate, rapid, and cost-effective system to apply as a routine method [11,12]. Furthermore, non-specific tests, such as serum biochemistry and hematology, are necessary to assess the extent of local infection and its systemic spread.

In the current case, the cat had a history of chronic wounds, subcutaneous nodules, and multiple draining tracts over a long period. No laboratory tests had been performed previously, and the lesions had progressively worsened. The cat had a history of fighting with another cat, suggesting that the organism might have been introduced through environmental contamination. Soil or water from the surrounding area could have penetrated the wound, leading to multiple punctured draining tracts and subcutaneous nodules. The differential diagnoses for cats presenting with cutaneous panniculitis, draining tracts, and subcutaneous nodules include deep fungal infections, bacterial infections, tissue foreign bodies, viral infections, and immune-mediated diseases [3,5,16]. In this case, cytology based on FNA of the subcutaneous nodule, stained with Wright–Giemsa, revealed pyogranulomatous inflammation. Acid-fast staining was not performed because it is not used routinely for skin cytology, which typically uses Diff-Quik and Wright–Giemsa stains. The diagnosis was confirmed through discharge culture, which revealed a *Mycobacterium abscessus* infection. In addition, antibiotic susceptibility testing was conducted to determine the appropriate treatment. Due to the challenging diagnosis of mycobacterial infections in cats, the disease is often overlooked [2,3,10], leading to frequent underdiagnosis [2]. It is important to consider mycobacterial infection as a differential diagnosis, given the potential zoonotic risk associated with human contact with infected cats [2,4,5,6,10]. Although rapidly growing non-tuberculous mycobacteria are environmental-opportunistic pathogens that are not typically transmitted from animals to humans [8], recent reports indicate an increasing incidence of NTM infections in humans [17], *Mycobacterium abscessus* is one of the most commonly identified species, responsible for severe respiratory, skin, and mucosal infections [4,17]. A strict hygiene regimen must be followed when in contact with an infected cat, and immunocompromised individuals should avoid contact with infected animals [2,4,17].

Treatment is almost always long-term and based on the susceptibility pattern of the species [2,8,17]. For example, for *Mycobacterium smegmatis* and *Mycobacterium fortuitum* complex, a fourth-generation fluoroquinolone (e.g., pradofloxacin or moxifloxacin) is recommended [2,5,8]. For *Mycobacterium chelonae-abscessus*, the first-line treatment choice is clarithromycin or azithromycin. Multidrug therapy with two or three drugs is recommended to improve treatment success and reduce the risk of relapse [2,5,10], with doxycycline being an appropriate choice for combination therapy. Although most patients show improvement within 1 month, treatment should continue for an additional 1–2 months after clinical resolution. For widespread or systemic infections, treatment should last 6–12 months [2,5,8,14], and in some cases, indefinite treatment may be required [5]. Because long-term treatment can be difficult for pet owners to maintain [10], unsuccessful treatment and the recurrence of lesions are common, and some cases ultimately lead to euthanasia [4,18]. Surgical excision of small cutaneous lesions should be considered; however, debulking larger lesions carries the risks of wound dehiscence and local recurrence of infection [2].

In the present case, successful treatment was achieved using a triple-drug therapy regimen. Initial treatment with oral azithromycin and topical amikacin gel, based on sensitivity testing, led to a partial decrease in the size and number of draining tracts, though some discharge remained. Subsequently, doxycycline was added to the regimen, resulting in complete wound healing. According to the literature, azithromycin, a macrolide antibiotic, can help manage granulomas primarily by acting as an anti-inflammatory agent. It may modulate the immune response within the granuloma, even when the granuloma is not caused by the bacteria that azithromycin directly targets. This means it can reduce inflammation and, in some cases, shrink the granuloma [19,20]. Similarly, doxycycline acts as an anti-inflammatory agent by inhibiting certain enzymes involved in granuloma formation and potentially modulating the immune response within the granuloma [21,22]. Prolonged use of azithromycin and doxycycline can cause gastrointestinal, hepatic, and renal effects; therefore, close monitoring of the physical condition, hematology, and blood chemistry profiles was conducted throughout the treatment period, and the cat exhibited normal signs, with no depression or vomiting reported. Mild anemia improved from 24.4 to 29.8. The serum biochemistry was within normal limits. No recurrent lesions were observed after 8 months, but regular and long-term monitoring is recommended to detect potential reinfection.

From the literature review, a case of *Mycobacterium abscessus* infection in a cat was reported in the Netherlands in 2009, presenting as localized pyogranulomatous dermatitis that progressed to larger dermal nodules. That case resulted in euthanasia, as elected by the owner [18]. Another case, reported in Switzerland in 2021, involved a highly resistant skin infection with phenotypic resistance to doxycycline, minocycline, and imipenem. Treatment with trimethoprim-sulfamethoxazole was unsuccessful, also leading to euthanasia [4]. In contrast, the present case represents the first successful treatment of *Mycobacterium abscessus* infection in a cat in Thailand and Southeast Asia, highlighting a major success in treating mycobacterial infections in cats by using appropriate antibiotics based on sensitivity testing results. It is hoped that this case treatment and description are of reference value in similar future cases of subcutaneous nodules or multiple sites of draining tract. Mycobacterial infection should be a differential diagnosis for appropriate treatment and prevention.

## 4. Conclusions

In this case, we report a *Mycobacterium abscessus* infection in a cat, identified using bacterial culture from discharge and the MALDI-TOF MS method. Successful treatment with complete wound healing was achieved using a triple-antibiotic regimen based on sensitivity testing. To date, there have been no reports of *M. abscessus* infection in cats in Thailand. This disease is easily underdiagnosed. Thus, in cats presenting with chronic subcutaneous nodules and multiple draining tracts, mycobacterial infection should be considered in the differential diagnosis. To the best of our knowledge, this is the first reported case in Thailand.

## Figures and Tables

**Figure 1 animals-15-00925-f001:**
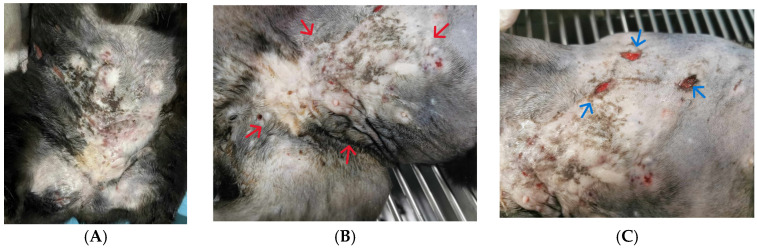
**Pre-diagnosis**. Skin lesions showing large area of subcutaneous nodules with multiple draining tracts on ventral abdomen. (**A**): Before clipping. (**B**): Left inguinal and prepubic areas with multiple, firm subcutaneous nodules and draining tract (red arrows). (**C**): Right ventral flank showing three opening sites of draining tract (blue arrows).

**Figure 2 animals-15-00925-f002:**
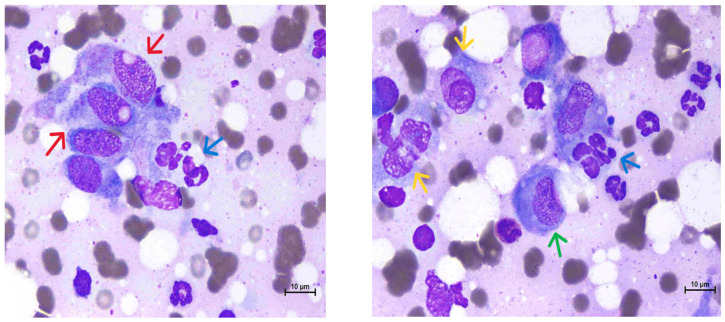
**Diagnosis**. Cytology from fine-needle aspiration of granuloma revealed multinucleated giant cells (yellow arrows), macrophages (green arrow), and large spindle cells (red arrows), as well as degenerate neutrophils (blue arrows), indicating pyogranulomatous inflammation. Wright–Giemsa staining. Magnification bars = 10 µm.

**Figure 3 animals-15-00925-f003:**
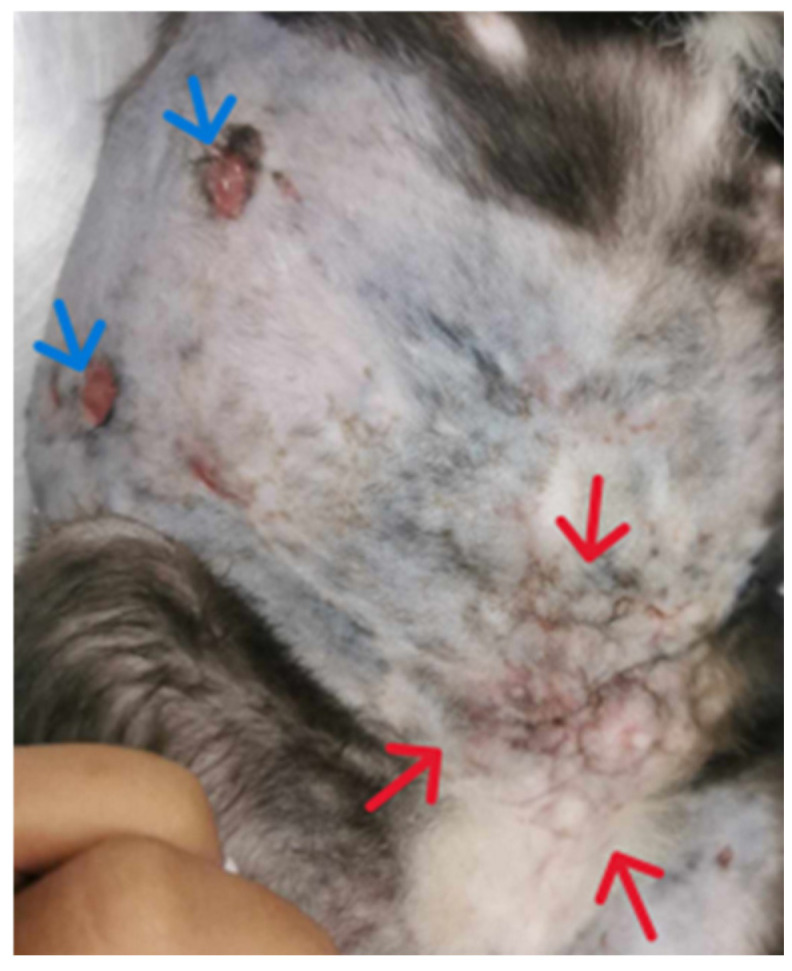
**Treatment and evolution**. Skin lesions after 36 days of azithromycin at 10 mg/kg PO q24h and topical amikacin gel: right flank showing two opening sites of draining tract (blue arrow), left inguinal and abdominal areas showing decreasing area of firm subcutaneous nodules (red arrows).

**Figure 4 animals-15-00925-f004:**
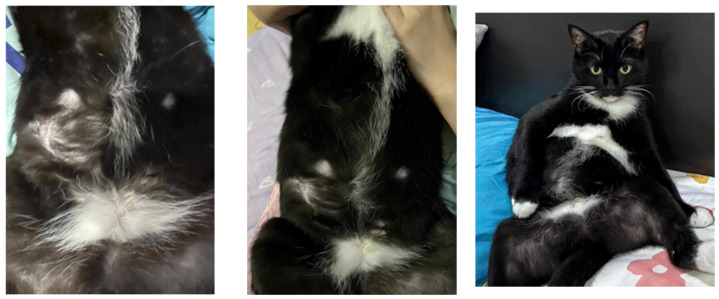
**Treatment and evolution**. Skin lesions showing complete improvement after 14 weeks of treatment, with no infection relapse.

**Table 1 animals-15-00925-t001:** Antimicrobial susceptibility report.

	Interpretation		
Antimicrobial	S	I	R
Amikacin	✓		
Azithromycin	✓		
Imipenem	✓		
Tetracycline	✓		
Doxycycline		✓	
Erythromycin		✓	
Amoxicillin–clavulanic acid			✓
Cefixime			✓
Cefovecin			✓
Ceftriaxone			✓
Cephalexin			✓
Chloramphenicol			✓
Clindamycin			✓
Gentamicin			✓
Meropenem			✓
Metronidazole			✓
Oxacillin			✓
Sulphamethoxazole/trimetroprim			✓
Ciprofloxacin			✓
Enrofloxacin			✓
Levofloxacin			✓
Marbofloxacin			✓
Pradofloxacin			✓

Abbreviations: S, susceptible; I, intermediate; R, resistant.

## Data Availability

The original contributions presented in this study are included in the article. Further inquiries can be directed to the corresponding author.

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
