# Peer review of "First Case Report of Successful Treatment of Mycobacterium abscessus Infection in a Cat in Thailand"

_animals, 2025, doi:10.3390/ani15070925_

Round 1
Reviewer 1 Report
Comments and Suggestions for Authors
As a preliminary work and report of a successful case of a problem that is rarely discussed, this article has value. However, it needs improvements, especially in the description of the case.
The case description definitely needs to be improved! I noticed some flaws, since the abstract mentions topical antibiotics in addition to azithromycin and the description was omitted. I would suggest that the text description be divided into pre-diagnosis (where the lesions are described and what type of drugs has already been given); diagnosis (where the diagnostic methodology is shown); and treatment and evolution.
and the categorization of the photos should be done according to this order. Please, reformulate this part in order to do justice to the clinical veterinary work done.
The discussion should also be improved. There are several flaws. It is important to note that this is the first report in Thailand, and it is clear that in other countries this infection has also been reported and discussed as being difficult to treat.
If it is a "common" but underdiagnosed infection, what happens to the undiagnosed cases? Do they end up developing unfavorably? Due to difficult treatment, is euthanasia chosen? Or does prolonged treatment with some commonly used antibiotic have positive effects? Please promote discussion in the presented article, otherwise this is just a clinical description.
In addition to all this, the conclusions should clearly show the "lesson" that can be learned from this case: - the importance of different diagnostic methods... - being meticulous... - using an antibiogram and applying the correct antibiotic... And exploring a little what preventive measures veterinarians can apply in these cases and what the warning signs are.
This work has the potential for discussion at the public health level and this point should be clear.
Comments on the Quality of English LanguagePlease, improve some parts.
Reviewer 2 Report
Comments and Suggestions for Authors
Upon reviewing the manuscript titled "First case report of successful treatment of Mycobacterium abscessus infection in cat in Thailand" I offer the following peer review comments:
1. The manuscript claims this is the first successful treatment of M. abscessus infection in a cat in Thailand. However, comparisons with similar cases in other countries (e.g., treatment protocols, resistance patterns, or epidemiological uniqueness) are lacking. Please elaborate on the distinctiveness of this case in terms of therapeutic approach, regional resistance profiles, or public health implications.
2. The initial combination of azithromycin and topical amikacin was based on susceptibility testing. However, why was doxycycline added at day 36? Were there clinical or laboratory indicators necessitating this change? Clarify the evidence supporting this modification.
3. Prolonged use of azithromycin and doxycycline may cause gastrointestinal, hepatic, or renal adverse effects. The manuscript does not mention monitoring of these parameters. Provide data on toxicity monitoring or justify its absence.
4. Is an 8-month follow-up sufficient to evaluate relapse risk for M. abscessus in cats? Discuss the pathogen’s latency characteristics and the need for longer-term surveillance.
5. The infection source was hypothesized to be a stray cat fight, but environmental testing (e.g., soil/water for mycobacteria) was not performed. Was environmental contamination considered? Expand the discussion on potential exposure routes.
6. Suspected typographical errors in Table 1 (e.g., “Centamycin” likely refers to Gentamicin; “Burofloxacin” may be Balofloxacin). Verify and correct all antimicrobial names to ensure scientific accuracy.
7. Figures 1–3 lack scale bars or quantitative descriptions (e.g., lesion size, number of draining tracts). Add annotations or metrics to objectively assess disease progression.
8. Recent regional studies on feline mycobacteriosis (e.g., from other Asian countries) and post-2020 advancements in the Journal of Feline Medicine and Surgery are not cited. Update references and strengthen the regional context.
A comprehensive revision is required to enhance scientific rigor. Resubmission after addressing these points is recommended.
Reviewer 3 Report
Comments and Suggestions for Authors
The manuscript "First case report of successful treatment of Mycobacterium abscessus infection in cat in Thailand" is an interesting case report, however, there is very little information to support the results. I recommend the authors to pay attention to the following observations and comments.
It is necessary to add the origin of the medicines, Laboratory, and country.
Granulomas have a large, well-organized capsule of connective tissue. How do you justify that the antibiotic penetrated and cured these lesions?
Why didn't they remove them and perform a histopathological study?
Could the authors provide more details on how the identification of M. abcesus by MALDITOF-SS was performed? Why was no acid-fast bacilli stain performed?
Authors are required to add data from clinical analyses, blood chemistry, hematology, etc.
Authors need to add the cytology image.
Round 2
Reviewer 1 Report
Comments and Suggestions for Authors
Dear authors,
I think that after the improvements made, the case is clearer and with a substantial increase in interest.
I suggest its publication.
Congrats.
Reviewer 2 Report
Comments and Suggestions for Authors
Authors have revised the manuscript and it could be published.
Reviewer 3 Report
Comments and Suggestions for Authors
The authors answered all my questions, I have no further comments for this manuscript.